# 2,2-Diphenethyl Isothiocyanate Enhances Topoisomerase Inhibitor-Induced Cell Death and Suppresses Multi-Drug Resistance 1 in Breast Cancer Cells

**DOI:** 10.3390/cancers15030928

**Published:** 2023-02-01

**Authors:** Monika Aggarwal

**Affiliations:** Department of Oncology, Lombardi Comprehensive Cancer Center, Georgetown University, Washington, DC 20007, USA; ma1274@georgetown.edu; Tel.: +1-202-687-3648; Fax: +1-202-687-1068

**Keywords:** isothiocyanate, p53 mutant, reactivation, chemoresistance, MDR1, gain-of-function, chemotherapy

## Abstract

**Simple Summary:**

Mutations in the p53 gene frequently occur in human cancers. The mutations are localized to “hotspot” residues and are classified as structural or contact mutants. p53 mutations disable wild-type (WT) p53 activity; they exert a “dominant negative” effect on WT p53 activity or a “gain-of-function” activity rendering the p53 mutants oncogenic. Mutant p53 rescue by small molecules is a promising chemotherapeutic strategy. Previously, we discovered that a dietary-related phenethyl isothiocyanate could reactivate p53^R175H^ structural mutants in HER2 + SK-BR-3 breast cancer (BC) cells. The current study revealed that 2,2-diphenethyl isothiocyanate (DPEITC) is a more potent synthetic analog that can inhibit the growth of different subtypes of BC cells, irrespective of p53 mutant-type, via mutant p53 rescue. DPEITC acts synergistically with the topoisomerase inhibitors doxorubicin and camptothecin. Furthermore, for the first time, we report that DPEITC suppresses multi-drug resistance 1 and ETS1, which have been shown to play a role in chemoresistance.

**Abstract:**

We previously reported that phenethyl isothiocyanate (PEITC), a dietary-related compound, can rescue mutant p53. A structure–activity relationships study showed that the synthetic analog 2,2-diphenylethyl isothiocyanate (DPEITC) is a more potent inducer of apoptosis than natural or synthetic ITCs. Here, we showed that DPEITC inhibited the growth of triple-negative breast cancer cells (MDA-MB-231, MDA-MB-468, and Hs578T) expressing “hotspot” p53 mutants, structural (p53^R280K^, p53^R273H^) or contact (p53^V157F^), at IC_50_ values significantly lower than PEITC. DPEITC inhibited the growth of HER2+ (p53^R175H^ SK-BR-3, p53^R175H^ AU565) and Luminal A (p53^L194F^ T47D) breast cancer (BC) cells harboring a p53 structural mutant. DPEITC induced apoptosis, irrespective of BC subtypes, by rescuing p53 mutants. Accordingly, the rescued p53 mutants induced apoptosis by activating canonical WT p53 targets and delaying the cell cycle. DPEITC acted synergistically with doxorubicin and camptothecin to inhibit proliferation and induce apoptosis. Under these conditions, DPEITC delayed BC cells in the G1 phase, activated p53 canonical targets, and enhanced pS1981-ATM. DPEITC reduced the expression of MDR1 and ETS1. These findings are the first report of synergism between a synthetic ITC and a chemotherapy drug via mutant p53 rescue. Furthermore, our data demonstrate that ITCs suppress the expression of cellular proteins that play a role in chemoresistance.

## 1. Introduction

Mutations in the p53 gene occur in a wide array of human cancers (www-p53.iarc.fr, accessed on 12 July 2022), predominantly through missense mutations localized to six “hotspot” residues which can be classified as either contact mutants (which disrupt DNA binding activity, e.g., R273H) or structural mutants (hallmarked by altered conformation, e.g., R175H). p53 mutations disable wild-type (WT) p53 activity; they exert a “dominant negative” effect on WT p53 activity or exhibit novel “gain-of-function” (GOF) activities rendering the mutants oncogenic [1,2,3,4]. Accumulating evidence has suggested that as a GOF activity, mutant p53 proteins form aberrant complexes with other proteins, such as transcription factors (e.g., ETS1) or others not directly related to gene transcription (e.g., MRE11 nuclease subunit of the MRE11/RAD50/NBS1 (MRN) complex). These protein interactions activate compensatory genes/pathways allowing tumor cell survival or altering signaling pathways regulated by WT p53 [5]. Mutant p53 interacts directly with ETS1 and upregulates the expression of multi-drug resistance 1 (MDR1), which plays a role in chemoresistance [6]. Mutant p53 interacts with the nuclease MRE11 and blocks recruitment of MRN complex to the site of a DNA double-strand break at the earliest step in homologous recombination and inhibits the activation of ataxia telangiectasia mutated (ATM) pathway [7]. Harnessing the mutant p53 status in cancer cells by small molecules to rescue WT p53 function is a potential therapeutic strategy. Synthetic small molecules from chemical libraries that rescue p53 point mutants to become transcriptionally competent are now under clinical investigation [8,9,10]. However, studies exploring the potential of natural compounds in targeting mutant p53 are scarce.

Phenethyl isothiocyanate (PEITC), abundant in watercress, exerts chemopreventive cancer effects in animal models, and epidemiological studies strongly support the role of dietary ITCs in protection against cancer in humans [11,12,13]. PEITC has been studied in multiple clinical trials [14]. Several activities have been proposed for PEITC, including the inhibition of cytochrome P450s, induction of phase II detoxifying enzymes, cell cycle arrest, oxidative stress, and its ability to bind and modify proteins [15,16,17,18,19]. Recently, we discovered a new anti-cancer mechanism for PEITC in which it rescues a p53 mutant in vitro as well as in breast (p53^R175H^ SK-BR-3) and prostate (p53^R175H^ LAPC-4 and p53^P223L/V274F^ DU145) cancer xenograft mouse models, thereby inhibiting tumor growth [20,21]. These studies provide the first evidence supporting PEITC’s potential as a “basket trials” agent to treat cancers harboring a p53 mutant, irrespective of cancer type.

A previous structure–activity relationships (SARs) study showed that an increase in the lipophilicity of arylalkyl ITCs by incorporating an aromatic ring is one of the key structural features important for their potency to deplete mutant p53 and induce apoptosis in human cancer cells [22]. Consistent with this, 2,2-diphenylethyl isothiocyanate (DPEITC) depletes mutant p53 to the greatest extent and is the most potent inducer of apoptosis of synthetic arylalkyl and naturally occurring ITCs examined previously in human cancer cells, including PEITC [22]. However, the mechanism(s) and molecular targets of DPEITC are not known. Here, we describe a study in which we have examined the effects of an additional aromatic ring on the potency of DPEITC and its ability to selectively target mutant p53, alone and in combination with chemotherapeutic drugs (topoisomerase inhibitors), in breast cancer, including human epidermal growth factor receptor 2 (Her-2/neu) (HER2+), Luminal A, and triple-negative breast cancer (TNBC). p53 is the most frequently mutated gene in TNBC, with a frequency of occurrence of ~88%, suggesting a critical role in TNBC. Due to the lack of estrogen receptor (ER), progesterone receptor (PR), and HER2, TNBC is not amenable to endocrine therapy and often shows resistance to chemotherapeutic agents currently used in the clinic [12,23,24]. Furthermore, we examined the effects of DPEITC on the expression of MDR1, which has been shown to mediate resistance to various cytotoxic drugs, including topoisomerase inhibitors [25], in mutant p53 TNBC cells. The lack of therapeutic options strongly rationalizes the need to investigate the potential role of mutant p53 as a therapeutic target in TNBC.

## 2. Materials and Methods

### 2.1. Cell Lines and Chemicals

All cell lines except Hs578T were cultured in a Roswell Park Memorial Institute (RPMI) 1640 medium (#10-040-CV) supplemented with fetal bovine serum (FBS, 10%) (#35-010-CV), penicillin/streptomycin (P/S) (1%) (#30-002-CI), and L-glutamine (1%) (#25-005-Cl). Hs578T cells were cultured in Dulbecco’s modified Eagle medium (DMEM) (#10-013-CV) containing FBS, P/S, L-glutamine at the concentrations listed above, and with 0.25% insulin (#12585-014, Gibco, Fisher Scientific, Waltham, MA, USA). All media and media supplements except insulin were purchased from VWR, Radnor, PA, USA. Cell lines were from Georgetown University, Washington, DC, Tissue Culture Source Resource. DPEITC was obtained from TransWorldChemicals, Rockville, MD, USA (Product ID: D6216). Mycoplasma contamination was not detected in any cell line.

### 2.2. Cell Proliferation Assays

The effect of DPEITC on cell proliferation was determined by the WST-1 assay (#5015944001, Roche, Indianapolis, IN, USA), as described previously [20,21]. Briefly, 4000 cells per well containing a desired concentration of DPEITC at 1% dimethyl sulfoxide (DMSO) were plated onto a 96-well microtiter plate in duplicate. As a control, 4000 cells containing 1% DMSO were seeded in duplicate. For background subtraction, wells containing the medium were used. Plates were incubated at 37 °C for 24 or 72 h, followed by the addition of the WST-1 reagent for 2 h. OD_450_ was measured using a microplate reader (Bio-Rad). A ratio of OD_450_ values for respective cells grown in the presence of DPEITC in comparison to the presence of DMSO was calculated to determine the percentage of cell proliferation. Similar assays were performed to determine the effects of DPEITC on cells transfected with p53 siRNA or non-specific (NS) siRNA described below.

To determine the effects of co-treatment with DPEITC and doxorubicin (#25316-40-9) or camptothecin (CPT) (#C-9911) on cell proliferation, cells were treated with the indicated concentrations of DPEITC, topoisomerase inhibitor, or both. Cell proliferation was then measured by performing WST-1 assays, as described previously. Combination index (CI) values were evaluated by isobolographic analysis [26]. Doxorubicin and CPT were purchased from Millipore Sigma, Rockville, MD, USA.

### 2.3. siRNA Transfection in Cells

The ON-TARGETplus human TP53 siRNA was obtained from SMARTPool (#L-003329-00-0005, Horizon Discover/PerkinElmer/Dharmacon, Cambridge, UK). The siRNA was transfected into cells using Lipofectamine 2000 transfection reagent (#11668019, Invitrogen, Thermofisher Scientific, Pittsburgh, PA, USA), as described previously [20], except the transfected cells were treated with DPEITC. Briefly, MDA-MB-231, MDA-MB-468, and Hs578T cells were plated to 50–60% confluence in 10 cm dishes 24 h before transfection. The siRNA (0.430 nmol) was mixed with 43 μL of Lipofectamine 2000 in 1 mL of Opti-MEM (#31985-070, Gibco, Invitrogen, Thermofisher Scientific, Pittsburgh, PA, USA). The mixture was added to the cells, which were subsequently incubated for 5 h. After 24 h, a second transfection was performed similarly. Seventy-two hours after the initial transfection, the cells were treated with DPEITC or DMSO (as a control) for cell proliferation or Western blotting, as required.

### 2.4. Annexin V Staining

Cells were harvested 72 h after treatment with the indicated concentration of DPEITC or DMSO, and Annexin-V staining was carried out in accordance with the manufacturer’s instructions (Biolegend, San Diego, CA, USA). Briefly, cells were harvested after the treatment, washed with PBS, and resuspended in 0.5 mL of Annexin V binding buffer. Cells were then collected by centrifugation, and 5 μL of the fluorescein isothiocyanate (FITC) fluorochrome-conjugated Annexin V were added to the residual buffer. The cell suspension was incubated at RT in the dark for 15 min, followed by the addition of 0.5 mL of Annexin V binding buffer and 5 μL of propidium iodide (PI) staining solution (0.1 μg/mL). Stained cells were analyzed for Annexin V positive and PI negative staining for early apoptosis with a BD LSRFORTESSA instrument (BD Biosciences, San Jose, CA, USA), as described previously [20,21].

### 2.5. Lysate Preparation and Western Blot Analysis

Cells were harvested after 3 h of treatment with DPEITC (at the indicated concentrations) or DMSO by centrifugation at 1600× *g* for 10 min at 4 °C and were washed once with PBS. Lysates were prepared using a lysis buffer containing a protease inhibitors cocktail, as described previously [20,21]. Briefly, cell suspensions in lysis buffer were incubated on ice for 30 min, and the supernatant was collected by centrifugation at 18,500× *g* for 10 min at 4 °C. Then, 50 μg (MDA-MB-231, MDA-MB-468, Hs578T, SK-BR-3, AU565, and T47D) or 100 μg (MCF7) of the supernatants were loaded on 4–12% SDS-PAGE. The PVDF blots after transfer were probed using primary p53 (DO-1) (#sc-126, Santa Cruz Biotechnology, Dallas, TX, USA) and GAPDH (#NB300-221, Novus Biologicals, Littleton, CO, USA) antibodies and secondary peroxidase-labeled anti-mouse IgG (1:1000, #170-6516, Bio-Rad, Hercules, CA, USA). The blots were developed using the ECL Prime Western Blot Detection Kit per the manufacturer’s instructions (#RPN2232, Amersham, GE Healthcare, Pittsburgh, PA, USA).

To determine the effects of Nutlin-3 (#N6287, Sigma, St. Louis, MO, USA) on the selective targeting of mutant p53 by DPEITC, cells were treated at the indicated concentrations of DMSO, DPEITC, Nutlin-3, or DPEITC and Nutlin-3 for 3 h. The levels of p53 in cell lysates were detected by Western blotting using p53 (DO-1) and GAPDH antibodies, as described above.

To detect expression levels of p53 mutants in MDA-MB-231, MDA-MB-468, and Hs578T cells transfected with p53 siRNA or NS siRNA, the cells were harvested after 72 h of initial transfection (day 1) and after 72 h of cell proliferation assay (day 4). The blots of cell lysate proteins were probed using p53 (DO-1) and GAPDH antibodies, as described above.

To detect ETS1 or MDR1 proteins, cells were treated with DMSO, DPEITC, or PEITC for 24 h. The blot with cell lysates was probed with primary (anti-ETS1 (1:500) (#sc-55581) or anti-MDR1 (1:1000) (#sc-55510) from Santa Cruz Biotechnology, Dallas, TX, USA) and secondary peroxidase-labeled anti-mouse IgG antibodies using the ECL Prime Western Blot Detection Kit, as described above.

To detect phosphorylation of ATM or activation of poly (ADP-ribose) polymerase-1 (PARP1), cells were treated with DMSO, DPEITC, 3-amino-1,2,4-triazole (ATZ), DPEITC and ATZ, CPT, or DPEITC and CPT, as indicated for 24 h. The PVDF membrane with the cell lysate protein was probed with primary anti-pATM Ser1981 antibody (1:500) (#sc-47739, Santa Cruz Biotechnology, Dallas, TX, USA), PARP1 (1:1000) (#614302, Biolegend, San Diego, CA, USA)), or anti-GAPDH antibody and secondary peroxidase-labeled anti-mouse IgG antibodies, respectively. The density of the bands in the treated samples relative to that of DMSO control was determined using Bio-Rad Image Lab software (Bio-Rad, Hercules, CA, USA). Original blots can be found at Appendix A.

### 2.6. Immunoprecipitation

MDA-MB-231, MDA-MB-468, and Hs578T cells were harvested after 4 h treatment with DPEITC or DMSO (as a control), and lysates were processed for immunoprecipitation, as described previously [20,21]. Briefly, 150 μg of the lysates were precleared at 4 °C for 30 min with protein G-agarose beads (#11719416001, Millipore Sigma, Rockville, MD, USA). The precleared lysates were then gently tumbled at 4 °C for 2 h with a mutant p53 conformation-specific mouse PAB240 antibody (#OP29, Calbiochem, Millipore Sigma, Rockville, MD, USA), followed by the addition of protein G-agarose beads and additional incubation for 2 h at 4 °C. The beads were washed three times with lysis buffer, and immunoprecipitated proteins eluted in Laemmli buffer were analyzed by Western blotting with a rabbit polyclonal FL393 (sc-6243, Santa Cruz Biotechnology, Dallas, TX, USA) as a primary antibody. For the secondary antibody, peroxidase-labeled mouse anti-rabbit IgG (1:5000, sc-2357, Santa Cruz Biotechnology, Dallas, TX, USA) was used. The blot was developed using the ECL Prime Western Blot Detection Kit according to the manufacturer’s protocol. As an input control, the blot was probed with p53 (DO-1) and then re-probed with an anti-GAPDH antibody.

### 2.7. Real-Time Quantitative Polymerase Chain Reaction (qRT-PCR) Assay

RNA was extracted from cells that were treated for 24 h with DMSO or DPEITC using a Qiagen RNeasy Kit (#74104, Qiagen, Valencia, CA, USA), as described previously [20,21]. cDNA was synthesized using a High Capacity RNA to cDNA Kit (#4387406, Applied Biosystems, Invitrogen, Thermofisher Scientific, Pittsburgh, PA, USA) and the gene expression level was measured by qRT-PCR using TaqMan gene expression assays, including p21 (#4331182, Hs00355782), NOXA (#4331182, Hs00560402), MDM2 (#4331182, Hs01066930), and BAX (#4331182, Hs00180269) and TaqMan fast advanced master mix (#4444556) (Applied Biosystems, Invitrogen, Thermofisher Scientific, Pittsburgh, PA, USA). For normalization, the expression level of GAPDH (#4331182, Hs03929097, Applied Biosystems, Invitrogen, Thermofisher Scientific, Pittsburgh, PA, USA) was measured. The average is presented with standard deviations from triplicates of repeated experiments. For combination treatments, cells were treated with DMSO, DPEITC, CPT, or DPEITC and CPT for 24 h, followed by RNA extraction and qRT-PCR, as described above.

### 2.8. Cell Cycle Analysis

The effect of DPEITC (72h) on cell cycle progression was determined by PI staining followed by analysis using a Becton Dickinson FACS sort (BD Biosciences, San Jose, CA, USA) and Mod Fit program (Verity Software House, Topsham, ME, USA), as described previously [20,21]. Briefly, cells were harvested after treatment by centrifugation at 190× *g* for 3 min at 4 °C, washed once with PBS, and stored as a suspension in 1 mL of 70% ethanol at –20 °C overnight. Cells were then harvested by centrifugation at 420× *g* for 10 min and resuspended in 1 mL of freshly prepared PI staining solution (PBS with 0.1% Triton X-100, 0.05 μg/mL propidium iodide, 0.1 mg/mL RNase (Sigma)). The cell suspension was incubated at RT for 30 min in the dark, followed by incubation for 30 min at 4 °C before analysis. Similarly, cells were treated with DMSO, DPEITC, ATZ, DPEITC, and ATZ, CPT, or DPEITC and CPT for indicated periods of time and cell cycle analysis was performed, as described above.

### 2.9. Statistical Analysis

A two-tailed Student’s t-test was used to evaluate the statistical differences in canonical p53 targets, and *p*-values of ≤0.05 were considered significant. All statistical tests were two-sided.

## 3. Results

### 3.1. DPEITC Inhibits Growth and Induces Apoptosis in TNBC Cells with Different p53 Mutants, Contact or Structural

We examined the effects of DPEITC on the proliferation of TNBC cells expressing a p53 contact mutant (p53^R280K^ MDA-MB-231, p53^R273H^ MDA-MB-468) or a structural mutant (p53^V157F^ Hs578T). In these cancer cells, DPEITC reduced cell proliferation at lower concentrations (IC_50_ 4 µM) after 24 h (Appendix A) and IC_50_ ≤ 3 µM after 72 h (Figure 1A). Importantly, no significant inhibition of proliferation was detected for WT p53 MCF7 cells after 24 h, whereas treatment for 72 h resulted in an IC_50_ of 10 µM (Appendix A and Figure 1A). These results show that DPEITC inhibited proliferation in p53 mutant TNBC cells at IC_50_ that were ~3.3–5-fold lower than WT p53 MCF7 cells. Of note, DPEITC inhibited the proliferation of TNBC cells expressing p53 contact mutant (MDA-MB-231 and MDA-MB-468) at approximately three and four-fold lower concentrations compared to PEITC, after 24 and 72 h, respectively [27]. No significant difference was detected in the IC_50_ for Hs578T cells expressing p53^V157F^ structural mutant after 72 h of treatment with DPEITC (IC_50_ 4 µM) or PEITC (IC_50_ 4 µM); however, DPEITC exhibited approximately two-fold lower IC_50_ (4 µM) as compared to PEITC treated cells (IC_50_ 8 µM) after 24 h (Figure 1A and Appendix A). Collectively, these data suggest that DPEITC is a more potent inhibitor of proliferation than PEITC. To assess the apoptotic potential, we compared the percentage of Annexin V positive stained cells in DMSO or DPEITC (3 µM or 5 µM)-treated cells after 72 h. Annexin-V is a specific phosphatidylserine (PS) residue-binding protein found in the inner membrane of the cytoplasmic membrane but externalized during apoptosis. Hence, it can be used to detect early apoptotic cells (bottom right quadrant shown in the representative flow cytometry image). PI is a DNA-binding dye, and cells co-stained with Annexin V and PI indicate dead cells, later apoptotic or necrotic (upper right quadrant shown in the representative flow cytometry image). The live cells are not stained (bottom left quadrant shown in the representative flow cytometry image). The TNBC cells expressing p53 mutants (R280K, R273H, or V157F) showed a significant (~2.5–8.3-fold) increase in the percentage of Annexin V stained cells as compared to the control cells (DMSO treated) (bottom right quadrant shown in the representative flow cytometry image), whereas no significant apoptosis was detected for WT p53 MCF7 cells treated with DPEITC as compared to DMSO-treated cells (bottom right quadrant shown in the representative flow cytometry image) (Figure 1B and Appendix A). These results demonstrate that DPEITC preferentially inhibits the growth of TNBC cells harboring mutant p53.

To confirm the selective targeting of mutant p53 by DPEITC, we determined its anti-proliferative effects on p53-depleted TNBC cells. MDA-MB-231, MDA-MB-468, and Hs578T cells transfected with p53 siRNA showed ≥90% reduction in the level of p53^R280K^, p53^R273H^, and p53^V157F^ mutant proteins compared to NS siRNA transfected cells (Appendix A). p53 depletion in p53^R280K^ MDA-MB-231, p53^R273H^ MDA-MB-468, and p53^V157F^ Hs578T cells resulted in markedly reduced sensitivity to growth inhibition by DPEITC, whereas the cells transfected with NS siRNA remained highly sensitive (Appendix A). These results demonstrate that growth inhibition by DPEITC is at least partially dependent on mutant p53 status in TNBC cells.

### 3.2. DPEITC Depletes Mutant p53 Protein, but Not WT p53, in TNBC Cell Lines

A previous study showed that arylalkyl ITCs deplete mutant p53 [22]. Consistent with this, we showed that DPEITC depleted several p53 mutants, contact (R280K, R273H) and structural (V157F), in TNBC cells (Figure 1C). We further showed that there was no significant change in the expression level of WT p53 in MCF7 cells (Figure 1C). These results suggest that mutant p53 is a crucial target for DPEITC.

### 3.3. DPEITC Rescues Mutant p53, Induces a G2/M Phase Delay, and Activates ATM in Mutant p53 TNBC Cell Lines

Since DPEITC inhibited cell proliferation in a mutant p53-dependent manner and induced apoptosis in TNBC cells expressing mutant p53, we examined if the compound restores WT p53 functions to mutant p53. Firstly, we determined the effects of DPEITC on the conformation of p53 mutants by immunoprecipitation using a mutant p53 conformation-specific PAB240 antibody. Western blotting analysis showed a significant decrease in the immunoreactivity of p53^R280K^ MDA-MB-231, p53^R273H^ MDA-MB-468, and p53^V157F^ Hs578T cell lysates (Figure 2A). These results demonstrated that DPEITC could induce a conformational change in the three different p53 mutant proteins. Consistent with this, treatment with 5 µM DPEITC enhanced the expression of canonical p53 targets p21, BAX, and NOXA in MDA-MB-468 and Hs578T cells (Figure 2B). These results suggested that DPEITC restores WT p53 transactivation functions to p53 contact (p53^R273H^) as well as structural (p53^V157F^) mutants in MDA-MB-468 and Hs578T cells, respectively. DPEITC also upregulated the expression of p21, BAX, and NOXA in WT p53 MCF7 cells (Figure 2B). The WT p53-MDM2 autoregulatory feedback loop is crucial for regulating p53 activity [28]. The upregulation of MDM2 expression was detected in p53^R273H^ MDA-MB-468, p53^V157F^ Hs578T, and WT p53 MCF7 cells upon treatment with 5 μM DPEITC (Figure 2B), suggesting that DPEITC can rescue mutant p53 and also activates WT p53. Consistent with this idea, co-treatment of MDA-MB-231, MDA-MB-468, and Hs578T cells with DPEITC (3, 5, or 8 μM) and Nutlin-3 (10 μM), a specific MDM2 inhibitor, resulted in a significant accumulation of p53^R280K^, p53^R273H^, and p53^V157F^ proteins, respectively, as compared with respective cells treated with DPEITC alone (3, 5, or 8 μM, respectively) (Appendix A). These results demonstrate that DPEITC affects the stability of distinct rescued mutant p53 proteins, with varied potency, in TNBC cells. Furthermore, the reduced stability of p53^R280K^, p53^R273H^, and p53^V157F^ proteins is due to proteasomal degradation.

Because WT p53 plays a crucial role in cell cycle progression, we examined the effects of p53^R273H^ and p53^V157F^ reactivation to “WT-like” p53 functions on cell cycle progression in MDA-MB-468 and Hs578T cells, respectively. p53^R273H^ MDA-MB-468 cells treated with 3 μM DPEITC displayed a significant delay in the G2/M phase compared to DMSO-treated cells at 72 h (43.09% and 23.79%) (Figure 2C and Appendix A). DPEITC-treated Hs578T cells expressing a p53^V157F^ structural mutant also displayed delays in S- and G2/M phases compared to DMSO-treated cells (32.1% vs. 24.8% or 23.70% vs. 14.90%, respectively) (Figure 2C and Appendix A). MCF7 (WT p53) cells treated with DPEITC (3 μM) showed an increase in S- and G2/M phases compared with DMSO-treated cells (Figure 2C and Appendix A). These results confirmed that DPEITC restores “WT-like” functions to a contact (p53^R273H^) and a structural (p53^V157F^) p53 mutant in TNBC cells.

To substantiate that the changes in cell cycle progression were due to DPEITC-induced “WT-like” p53 functions to mutant p53 and not a general response to cellular stress, we examined the effects of stress induced by ATZ, an inhibitor of endogenous catalase, on the cell cycle. p53^R273H^ MDA-MB-468 cells treated with 2 mM ATZ did not show any significant change in the cell cycle compared to the DMSO-treated cells (31.5% vs. 30.5%, respectively), whereas cells treated with 3 μM DPEITC alone or in combination with 2 mM ATZ displayed a significant delay in the G2/M phase at 72 h (57% and 62.4%, respectively) (Figure 3A). p53^R280K^ MDA-MB-231 cells treated with 3 μM DPEITC alone or in combination with ATZ displayed delayed progression in the G2/M phase compared to DMSO-treated cells (25.9% and 26.2% vs. 17.7%, respectively), whereas no significant change was detected for cells treated with ATZ (2 mM) alone (17.9%) (Figure 3B). Consistent with this, MCF7 (WT p53) cells treated with DPEITC (3 μM) alone or 3 μM DPEITC in combination with 2 mM ATZ showed an increase in S- and G2/M phases compared with DMSO-treated cells, but no significant change in cells was observed upon treatment with 2 mM ATZ alone (Figure 3C). These results demonstrate that DPEITC affects the cell cycle specifically by restoring “WT-like” p53 functions to p53 mutants. In support of this, we also showed that DPEITC alone did not induce any significant delay in the cell cycle in p53 null MDA-MB-436 cells (Appendix A).

We then examined if the delay in the cell cycle was due to the activation of the DNA damage response (DDR) regulated by the ATM serine/threonine protein kinase pathway in the presence of rescued mutant p53. Previously, we showed that PEITC-induced rescue of WT p53 functions to mutant p53 abolish p53 mutant GOF activity to inhibit the activation of ATM [7,20,21]. p53^R280K^ MDA-MB-231 and p53^R273H^ MDA-MB-468 cells treated with 5 μM DPEITC showed autophosphorylation of ATM at S1981 compared to the DMSO control (Figure 4). Furthermore, p53 mutant cells (MDA-MB-231 and MDA-MB-468) co-treated with 5 μM DPEITC and 2 mM ATZ showed enhanced autophosphorylation of ATM at S1981 as compared with cells treated with DPEITC alone (Figure 4). In contrast, cells treated with ATZ alone did not show any significant autophosphorylation of ATM (Figure 4). We did not detect any significant accumulation of pATM-S1981 in WT p53 MCF7 cells treated with 5 μM DPEITC as compared to DMSO-control; however, a mild accumulation of pATM-S1981 was detected in MCF7 co-treated with 5 μM DPEITC and 2 mM ATZ (Figure 4). Collectively, these results demonstrate that DPEITC reactivates WT p53 transactivation functions of p53 mutants (p53^R280K^ and p53^R273H^), induces a delay in cell cycle progression, and activates the DDR response in MDA-MB-231 and MDA-MB-468, respectively.

### 3.4. DPEITC Inhibits Growth, Induces Apoptosis, and Rescues p53 Structural Mutants in HER2+ and Luminal A Breast Cancer Cells

We next examined the effects of DPEITC on the proliferation of HER2+ (SK-BR-3, AU565) and Luminal A (T47D) breast cancer cell lines expressing a p53^R175H^ or p53^L194F^ structural mutant, respectively. In these cancer cells, DPEITC reduced cell proliferation at concentrations ≥ 12 µM after 24 h (Appendix A), whereas, after 72 h, the IC_50_ value was 4 µM (Figure 5A). DPEITC (3 µM or 5 µM) enhanced the percentage of Annexin V stained SK-BR-3, AU565, and T47D cells compared to the DMSO-treated cells (bottom right quadrant shown in the representative flow cytometry image), respectively, after 72 h, suggesting that the compound induced apoptosis (Figure 5B and Appendix A). Furthermore, DPEITC depleted p53^R175H^ and p53^L194F^ mutants in a dose-dependent manner (Figure 5C). We also showed that AU565 cells co-treated with DPEITC (3, 5, or 8 μM) and Nutlin-3 (10 μM) display a significant accumulation of p53^R175H^ protein compared to cells treated with DPEITC alone (3, 5, or 8 μM, respectively) (Appendix A).

To determine if DPEITC restores WT functions to p53^R175H^ and p53^V194F^ mutants, we examined the expression of canonical p53 targets p21, BAX, and NOXA in SK-BR-3 and T47D cells, respectively, after treatment with 5 µM DPEITC. DPEITC upregulated expression of p21, BAX, and NOXA in SK-BR-3 and T47D cells (Figure 6A), demonstrating that DPEITC restores WT p53 transactivation functions to the p53 structural mutants in HER2+ and Luminal A breast cancer cells. Consistent with this idea, DPEITC induced G2/M phase arrest in p53^R175H^ SK-BR-3 and p53^V194F^ T47D cells (Figure 6B and Appendix A). Collectively, these results suggest that DPEITC rescues different mutant p53 types, contact and structural, irrespective of the breast cancer subtypes.

### 3.5. DPEITC Increases Cellular Sensitivity to Doxorubicin

Doxorubicin causes DNA damage by intercalating between the DNA base pairs and inhibiting the topoisomerase 2 enzyme [29]. Overexpression of mutant p53 is often associated with resistance to doxorubicin [30]. Therefore, we sought to assess if, in the presence of rescued mutant p53, DPEITC acts synergistically with doxorubicin to inhibit mutant p53 TNBC cell proliferation. TNBC cells expressing a p53 contact mutant (p53^R280K^ MDA-MB-231, p53^R273H^ MDA-MB-468), a structural mutant (p53^V157F^ Hs578T), or WT p53 (MCF7) were treated with doxorubicin (0.25 μM), DPEITC (0.25 μM), or both. Co-treatment of mutant p53 TNBC cells with DPEITC and the topoisomerase 2 inhibitor resulted in a ≥50% reduction in MDA-MB-231, MDA-MB-468, and Hs578T cell proliferation with combination index (CI) values of ≤0.8 (Figure 7A). Combination index values of <1 indicate synergy [26]. We did not detect any significant difference in the proliferation of MCF7 cells that were co-treated with DPEITC and doxorubicin compared with cells treated with DPEITC or doxorubicin alone (Appendix A). These results demonstrate that DPEITC and doxorubicin act synergistically to inhibit cell proliferation in mutant p53 TNBC.

### 3.6. Synergistic Effects of DPEITC and Camptothecin on Proliferation and Apoptosis Induction

To assess the effects of DPEITC on the ability of cells to cope with the DNA damage induced by CPT, TNBC cells expressing a p53 contact (p53^R280K^ MDA-MB-231, p53^R273H^ MDA-MB-468) or a structural (p53^V157F^ Hs578T) mutant were exposed to sublethal concentrations of topoisomerase 1 inhibitor (0.25 μM) and DPEITC (0.06 μM). Results from cell proliferation and cell death assays were compared in cells exposed to CPT or DPEITC alone. Co-treatment of mutant p53 TNBC cells with DPEITC and CPT resulted in a ≥50% reduction in cell proliferation, whereas neither compound alone could inhibit growth by ≥30% (Figure 7B), suggesting that the topoisomerase inhibitor and DPEITC act synergistically to inhibit cell proliferation. CI values of ≤0.8 further confirmed the synergism between DPEITC and CPT treatment (Figure 7B). We did not detect any significant synergism between 0.06 μM DPEITC and 0.25 μM CPT in WT p53 MCF7 cells (Appendix A).

As a hallmark of apoptosis, caspase 3 induces the cleavage of PARP1 [31]. Therefore, to determine the apoptotic potential of combined treatment, we examined the effects of CPT (0.25 μM) and DPEITC (0.06 μM) on PARP1 levels in p53^R280K^ MDA-MB-231 and p53^R273H^ MDA-MB-468 cells. Western blot analysis showed a significant decrease in the levels of PARP1 in p53^R280K^ MDA-MB-231 and p53^R273H^ MDA-MB-468 cells co-treated with CPT (0.25 μM) and DPEITC (0.06 μM) compared to untreated control cells and cells treated with either agent alone (Figure 7C). These results demonstrate an apparent synergistic effect of DPEITC and CPT on apoptosis induction.

### 3.7. Synergistic Effects of DPEITC and Camptothecin on Cell Cycle Progression, ATM Activation, and Transactivation Functions to p53 Mutants

Because co-treatment of mutant p53 TNBC cells with DPEITC and CPT induced apoptosis, we determined if DPEITC could restore WT p53 functions under these conditions. Therefore, we examined the effects of CPT (0.25 μM) and DPEITC (0.06 μM) co-treatment on cell cycle progression in p53^R273H^ MDA-MB-468 and p53^R280K^ MDA-MB-231 cells. Results from cell cycle analysis were compared in cells that were either untreated (DMSO control) or treated with CPT or DPEITC alone. p53^R280K^ MDA-MB-468 cells co-treated with DPEITC and CPT displayed a significant increase in G1- and S-phase cells at 24 h when compared to untreated cells (Figure 8A and Appendix A), suggesting that co-treatment with DPEITC and CPT inhibited cell proliferation by delaying cells not only in the G1 phase but also in the S phase. No significant change in cell cycle progression was observed in cells treated with DPEITC alone, whereas treatment with CPT alone induced S-phase delay (Figure 8A and Appendix A). Similarly, p53^R280K^ MDA-MB-231 co-treated with DPEITC and CPT displayed an increase in the percentage of cells in the G1 phase compared to cells treated with either agent alone (Figure 8A and Appendix A).

We next examined if the delay in the cell cycle was due to activation of the DDR, regulated by the ATM pathway. Here, we showed that co-treatment of p53^R273H^ MDA-MB-468 and p53^R280K^ MDA-MB-231 with DPEITC and CPT induced phosphorylation of ATM, whereas no significant activation of ATM was detected in cells treated with DPEITC or CPT alone (Figure 8B). Consistent with this, co-treatment with DPEITC and CPT enhanced the expression of canonical targets, p21 and BAX, respectively, compared to cells treated with either agent alone (Figure 8C,D). These results suggest that DPEITC induced rescue of mutant p53 in combination with CPT, leading to enhanced sensitivity.

### 3.8. DPEITC Treatment Suppresses the Expression of ETS1 and MDR1 Proteins

Energy-dependent drug efflux by ABC-transporters, including ABCB1/MDR1 (multi-drug resistance 1)/P-glycoprotein (P-gp), constitutes one of the best-characterized mechanisms of multi-drug resistance in vitro [32]. As a GOF activity, mutant p53 protein selectively upregulates the expression of MDR1 via its interaction with a transcription factor ETS1 [33]. In contrast, WT p53 does not interact with ETS1 and abolishes MDR1 activity [6,32,33,34]. Because DPEITC rescued mutant p53, we examined whether the reactivation of p53^R280K^ and p53^R273H^ to ‘WT-like’ p53 abolishes its ability to upregulate the expression of ETS1 and MDR1 in MDA-MB-231 and MDA-MB-468 cells, respectively. Therefore, we examined the expression of ETS1 and MDR1 in MDA-MB-231 and MDA-MB-468 cells treated with DPEITC. Western blot analysis of MDA-MB-231 and MDA-MB-468 cells treated with DPEITC (5 μM) revealed a significant decrease in the expression of ETS1 and MDR1 when compared with DMSO-treated cells (Figure 9A). No significant change in the expression of ETS1 and MDR1 was observed in the WT p53 MCF7 cells that were untreated or treated with DPEITC (Figure 9A). Previously, we found that dietary-related PEITC rescues mutant p53 [20,21]. As a control, we showed that PEITC treatment reduced the expression of ETS1 and MDR1 in mutant p53 MDA-MB-231 and MDA-MB-468 but not WT p53 MCF7 cells (Figure 9B). These results suggest that restoration of ‘WT-like’ p53 functions to mutant p53 by an ITC, synthetic or naturally occurring dietary-related, could abolish mutant p53 GOF activity to induce upregulation of MDR1 expression in cancer cells.

## 4. Discussion

In this study, we demonstrate that DPEITC can inhibit the proliferation of TNBC cells expressing different p53 mutant types, contact (p53^R280K^ MDA-MB-231 and p53^R273H^ MDA-MB-468) and structural (p53^V157F^ Hs578T). Of note, the IC_50_ values for DPEITC (with two aromatic rings) are significantly lower than that of PEITC (one aromatic ring) for p53^R280K^ MDA-MB-231 and p53^R273H^ MDA-MB-468 cell lines, as shown previously [27] and for p53^V157F^ Hs578T in the present study. These results showed that DPEITC is a more potent inhibitor of proliferation. Previous studies showed that the lipophilicity of the side chain moiety of aromatic ITCs is a crucial determinant of their apoptotic potential [22]. DPEITC inhibited the proliferation of WT p53 MCF7 cells; however, it induced apoptosis selectively in mutant p53 TNBC cells and depleted mutant p53, but not WT. Collectively, these results showed that mutant p53 TNBC cells were more sensitive and supported mutant p53 as a target of DPEITC.

DPEITC restored WT p53 transactivation functions to p53^R280K^, p53^R273H^, and p53^V157F^ mutants in TNBC cells. In addition to inducing the expression of a proapoptotic ‘multi-domain’ Bcl-2 family member BAX, DPEITC upregulated the expression of a proapoptotic ‘BH3-only’ class of BCL-2 family member, NOXA, which is a direct transcriptional target of p53 and has been shown to promote p53-dependent apoptosis under DNA damage stress [35]. These results showed that DPEITC could rescue mutant p53 irrespective of the p53 mutant type, contact or structural. Consistent with this idea, DPEITC caused a delay in the G2/M- or S- and G2/M-phase delay in MDA-MB-468 and Hs578T cells, respectively. Furthermore, autophosphorylation of ATM at S1981 in DPEITC-treated MDA-MB-231 and MDA-MB-468 cells indicated induction of DDR. We also detected an upregulation of canonical p53 targets and a delay of cell cycle progression in G2/M- and S-phase in WT p53 MCF7 cells but without the induction of apoptosis and the change in expression of WT p53. Supporting this, we did not detect any significant phosphorylation of ATM in DPEITC-treated WT p53 cells. Previously, we showed that PEITC-induced oxidative stress activates DDR in the presence of “WT-like” mutant p53 resulting in the activation of rescued mutant p53 and induction of apoptosis. Further studies are required to understand the mechanism of DPEITC in WT p53 cells.

Reinforcing the idea that DPEITC can rescue different p53 mutants, we showed that DPEITC inhibited proliferation and induced apoptosis in HER2+ (SK-BR-3 and AU565) and Luminal A (T47D) breast cancer cells expressing different p53 structural mutants, p53^R175H^ or p53^L194F^. DPEITC restored WT p53 transactivation functions to p53^R175H^ and p53^L194F^ mutants and caused a cell cycle delay in S- and G2/M or G2/M phase in SK-BR-3 and T47D cells, respectively. Collectively, these results suggest that DPEITC acts irrespective of the breast cancer sub-type but in a p53 mutant-dependent manner. Supporting this, DPEITC also depleted p53^R175H^ and p53^L194F^ mutants in HER2+ and Luminal A breast cancer cells. Previous studies showed that ITCs can bind the cysteine residues in the DNA binding domain of the purified mutant p53 protein and that their ability to deplete p53 mutants in cells seems to correlate with their binding affinities [22]. The data on DPEITC reinforce a mechanism involving the binding of DPEITC to the mutant p53 protein via the ITC functional group. Further studies are needed to address how a cancer cell executes an intricate balance between DPEITC-induced reactivation of “WT-like” p53 functions of mutant p53 and its depletion.

The synergistic effects of DPEITC and topoisomerase inhibitors in mutant p53 TNBC expressing different p53 mutants, contact or structural, were apparent by the inhibition of proliferation at CI values ≤ 0.8. These drugs act by inducing DNA damage and require WT p53 to induce apoptosis. Indeed, co-treatment with DPEITC and CPT resulted in the induction of apoptosis in MDA-MB-231 and MDA-MB-468 cells. Delay in cell cycle progression in the G1 phase, and upregulation of p21 and BAX in MDA-MB-231 and MDA-MB-468 cells co-treated with DPEITC and CPT suggests that DPEITC restores “WT-like” p53-functions to p53 mutant in TNBC cells. We did not detect any apparent synergism between DPEITC and the topoisomerase inhibitors at the lower sublethal doses in WT p53 MCF7 cells. These results are consistent with the findings that cells expressing WT p53 protein were less sensitive to DPEITC as compared to the mutant p53 cells. Collectively, the results with WT p53 cells treated with DPEITC alone or in combination with a chemotherapy drug suggest that these cells may respond to DPEITC via a divergent mechanism than the mutant p53 cancer cells. We further showed that DPEITC and CPT co-treatment activated ATM. Mutant p53 impairs the cell’s ability to activate the ATM/CHK2 pathway in the presence of DNA DSB damage [7,20,21]. Thus, our results demonstrate that the GOF activity of mutant p53 to inhibit the activation of ATM in the presence of DNA DSB damage was abolished by DPEITC-induced rescue of mutant p53.

As a GOF activity, mutant p53 upregulates the expression of MDR1 via its interaction with ETS1 [6,32,33,34]. Interestingly, we showed that DPEITC treatment reduces the expression of MDR1 and the transcription factor ETS1. In cancer therapeutics, there is a heightened interest in understanding the mechanisms that confer resistance to anti-cancer drugs, one of the major obstacles to efficacious chemotherapy. ATP-dependent drug transporters, including MDR1, breast cancer resistance protein (BCRP) or ABCG2, and multi-drug resistance-associated protein 1 (MRP1) or ABCC1, have been shown to be effective in lowering the intracellular concentration of drugs and the development of multi-drug resistance [36]. ETS1 is a transcription factor that regulates the transcription of MDR1 [6]. This study provides the first evidence of the restoration of the WT p53 functions to a p53 mutant can abolish mutant p53 GOF activity to upregulate the expression of ATP-dependent drug efflux transporter MDR1 (ABCB1/P-gp), which has been proposed to mediate resistance to various cytotoxic drugs, including doxorubicin [32,36]. More studies are required to understand how DPEITC-induced mutant p53 rescue affects the activities of ETS1 and MDR1 and may help to sensitize anti-cancer drug-resistant cells to chemotherapy drugs. Nevertheless, a new insight is afforded by our study that an isothiocyanate can rescue mutant p53 in combination with a chemotherapy drug at a sublethal dose to induce synthetic lethality. These studies lay a platform for the future development of novel combination strategies using ITCs in TNBC. Furthermore, our findings open avenues to investigate the potential of mutant p53 restoration for overcoming drug resistance.

## 5. Conclusions

In summary, this is the first report that DPEITC, a synthetic ITC, inhibits the growth of TNBC cancer cells harboring different “hotspot” p53 mutants (structural and contact) via mutant p53 rescue and is a more potent inhibitor than the naturally occurring dietary-related PEITC. We also showed that DPEITC inhibits the growth of HER2+ and Luminal A breast cancer sub-types harboring different p53 contact mutants and via mutant p53 rescue. Collectively, these results suggest that the anti-cancer activity of DPEITC is p53 mutant type-dependent and breast cancer subtype-independent. DPEITC act synergistically with topoisomerase inhibitors to inhibit the growth of mutant p53 TNBC. Importantly, DPEITC rescues the p53 mutant at a sub-lethal dose with a chemotherapy drug. Furthermore, our finding that DPEITC could reduce the expression of ETS1 and MDR1 proteins selectively in mutant p53 TNBC cells, and not WT, opens up opportunities to investigate small molecules induced mutant p53 restoration as one of the potential strategies to overcome chemoresistance.

## Figures and Tables

**Figure 1 cancers-15-00928-f001:**
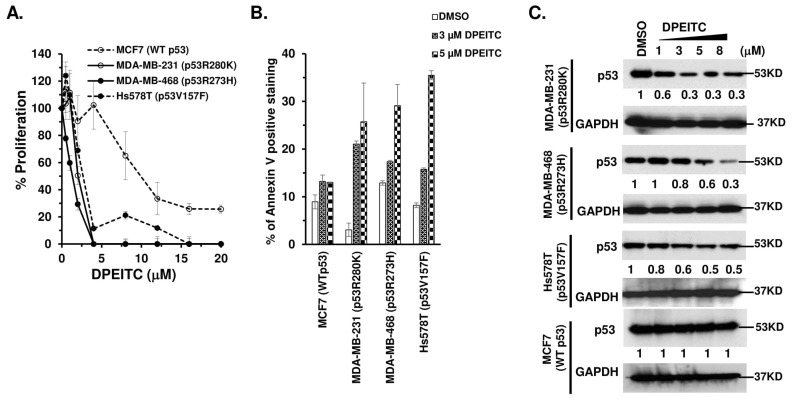
DPEITC inhibits cell proliferation, induces apoptosis, and affects mutant p53 expression levels in TNBC cell lines with mutated p53. MCF7 (WT p53), MDA-MB-231 (p53^R280K^), MDA-MB-468 (p53^R273H^), and Hs578T (p53^V157F^) cells were treated with DMSO or DPEITC for 72 h and analyzed for percent cell proliferation (**A**) or apoptosis (**B**). (**C**) Cell lysates of TNBC cell lines with WT p53 (MCF7) or mutant p53 (MDA-MB-231, MDA-MB-468, Hs578T) treated with DMSO or the indicated concentration of DPEITC for 3 h were analyzed by Western blotting. Experiments were performed in triplicate. Error bars represent SD.

**Figure 2 cancers-15-00928-f002:**
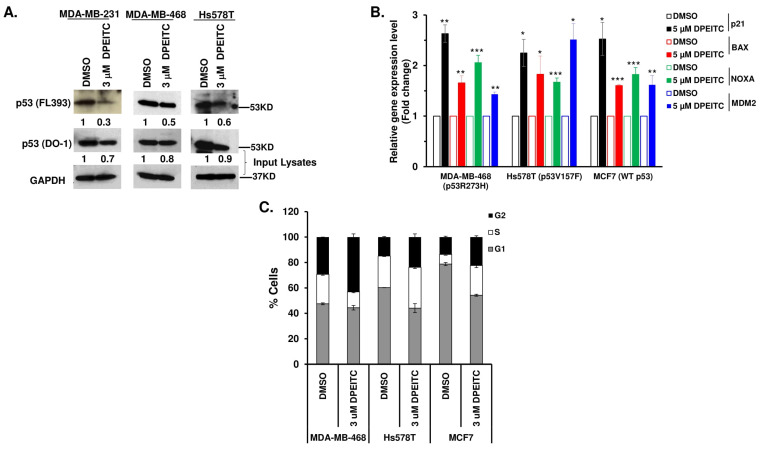
DPEITC induces a conformational change in p53 mutants, restores p53 mutant proteins’ transactivational functions, and induces G2 phase arrest in TNBC cell lines. (**A**) Immunoprecipitation to determine the effects of DPEITC on the conformation of the p53 mutant proteins, p53^R280K^, p53^R273H^, and p53^V157F^ from MDA-MB-231, MDA-MB-468, and Hs578T cell lysates using a mutant p53 conformation-specific PAB240 antibody. The immunoprecipitated proteins were analyzed by Western blotting with p53 (FL393) antibody. The Western blot image for p53 (DO-1) serves as an input control. The blot was re-probed with an anti-GAPDH antibody. (**B**) MCF7 (WT p53), MDA-MB-468 (p53^R273H^), and Hs578T (p53^V157F^) cells were treated with DMSO or 5 μM DPEITC for 3 h and qRT-PCR of p53 canonical target genes was performed. (*** *p* ≤ 0.0001, ** *p* ≤ 0.001 and * *p* ≤ 0.04). (**C**) MCF7 (WT p53), MDA-MB-468 (p53^R273H^), and Hs578T (p53^V157F^) cells were treated with DMSO or 3 μM of DPEITC for 72 h and analyzed by flow cytometry. Experiments were performed in triplicate. Error bars represent SD.

**Figure 3 cancers-15-00928-f003:**
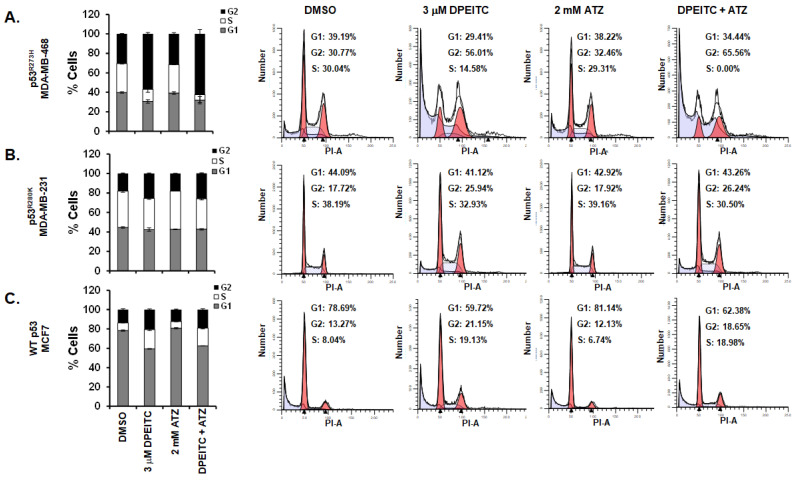
Effects of DPEITC and ATZ co-treatment on cell cycle progression in TNBC cell lines. p53^R280K^ MDA-MB-231 (**A**), p53^R273H^ MDA-MB-468 (**B**), and WT p53 MCF7 (**C**) cells were treated with DMSO, 3 μM DPEITC, 2 mM ATZ, or 3 μM DPEITC and 2 mM ATZ for 72 h and analyzed by flow cytometry. Experiments were performed in triplicate. Error bars represent SD. Representative cell cycle images are shown.

**Figure 4 cancers-15-00928-f004:**
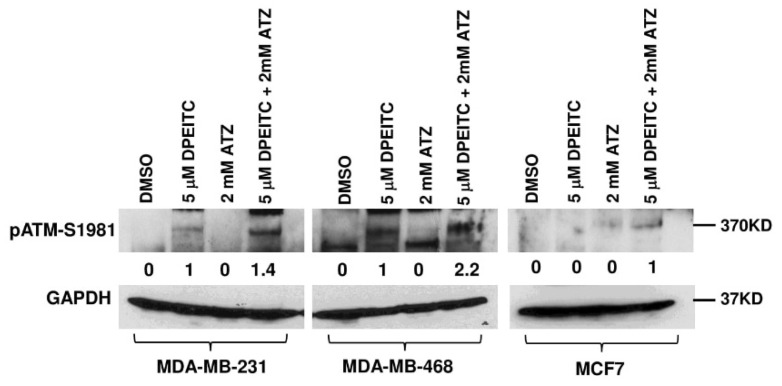
Effects of DPEITC and ATZ co-treatment on activation of ATM in mutant p53 TNBC cells. MDA-MB-231 (p53^R280K^), MDA-MB-468 (p53^R273H^), or MCF7 (WT p53) cells were treated with DMSO, 5 μM DPEITC, 2 mM ATZ, or both for 24 h. Blots were probed using the anti-pATM S1981 antibody and re-probed with the GAPDH antibody. Experiments were performed in triplicate.

**Figure 5 cancers-15-00928-f005:**
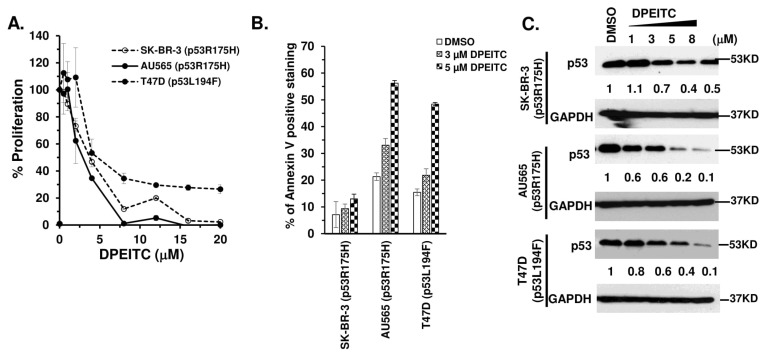
DPEITC inhibits cell proliferation, induces apoptosis, and affects p53 mutant expression levels in HER2_+_ and Luminal A breast cancer cell lines. HER2+ SK-BR-3 (p53^R175H^), AU565 (p53^R175H^), and Luminal A T47D (p53^L194F^) cell lines were treated with DMSO or DPEITC for 72 h and evaluated for percent cell proliferation (**A**) or apoptosis (**B**). (**C**) Cell lysates of HER2+ and Luminal A breast cancer cell lines treated with DMSO or the indicated concentration of DPEITC for 3 h were analyzed by Western blotting. Experiments were performed in triplicate. Error bars represent SD.

**Figure 6 cancers-15-00928-f006:**
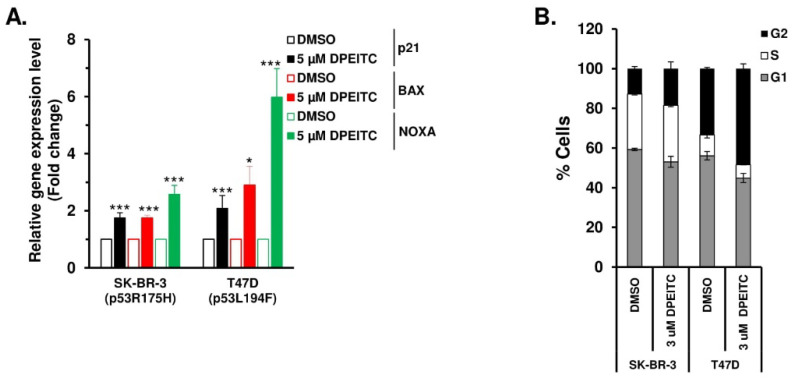
DPEITC reactivates different p53 mutant proteins’ transactivational functions and induces G2 phase arrest in HER2+ and Luminal A breast cancer cell lines. (**A**) SK-BR-3 (p53^R175H^) and T47D (p53^L194F^) cells were treated with DMSO or 5 μM DPEITC for 3 h, and qRT-PCR of p53 regulated genes was performed. (*** *p* ≤ 0.0001 and * *p* ≤ 0.03). (**B**) SK-BR-3 (p53^R175H^) and T47D (p53^L194F^) cells were treated with DMSO or 5 μM of DPEITC for 72 h and analyzed by flow cytometry. Experiments were performed in triplicate. Error bars represent SD.

**Figure 7 cancers-15-00928-f007:**
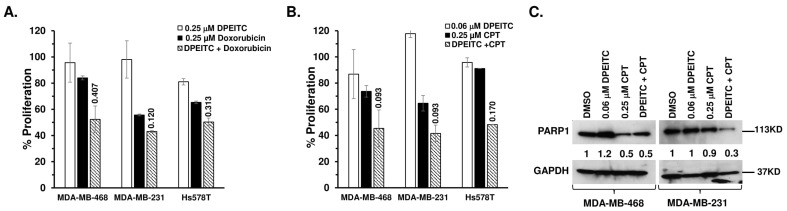
DPEITC enhances p53 mutant TNBC cell’s sensitivity to topoisomerase inhibitors. (**A**) MDA-MB-468 (p53^R273H^), MDA-MB-231 (p53^R280K^), or Hs578T (p53^V157F^) cells were treated with DPEITC, CPT, or both for 24 h. (**B**) MDA-MB-468 (p53^R273H^), MDA-MB-231 (p53^R280K^), or Hs578T (p53^V157F^) cells were treated with DPEITC, doxorubicin, or both for 72 h. Percent cell proliferation was determined by the WST-1 assay. Experiments were performed in triplicate. Error bars represent SD. (**C**) Effects of DPEITC on PARP1 levels. MDA-MB-468 (p53^R273H^) or MDA-MB-231 (p53^R280K^) cells were treated with DPEITC, CPT, or both for 24 h. In total, 100 μg of the cell lysate fractions were resolved by SDS-PAGE and probed with an anti-PARP antibody. Blots were stripped and re-probed with anti-GAPDH as a loading control. Experiments were performed in triplicate.

**Figure 8 cancers-15-00928-f008:**
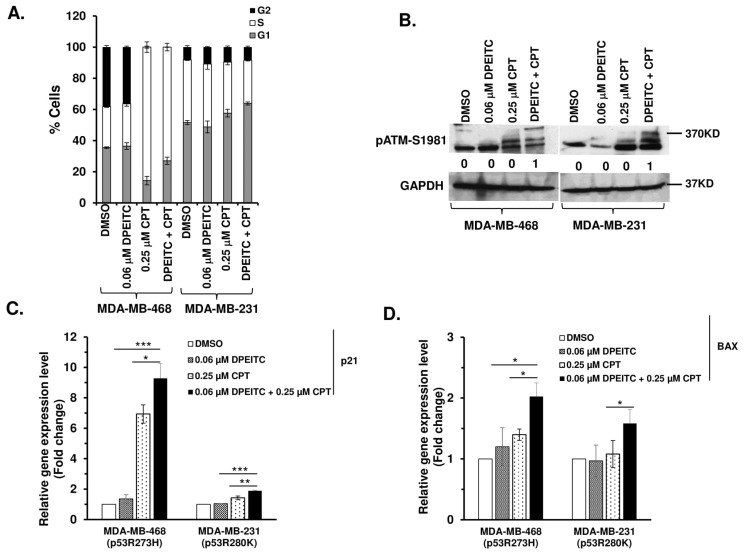
Effects of DPEITC and CPT co-treatment on cell cycle progression, activation of ATM, and canonical p53 targets. (**A**) MDA-MB-468 (p53^R273H^) or MDA-MB-231 (p53^R280K^) cells were treated with DPEITC, CPT, or both for 24 h and analyzed by flow cytometry. (**B**) MDA-MB-468 (p53^R273H^) or MDA-MB-231 (p53^R280K^) cells were treated with DPEITC, CPT, or both for 24 h. Blots were probed using the anti-pATM S1981 antibody and re-probed with the GAPDH antibody. (**C**) and (**D**) The expression of canonical p53 target genes p21 (**C**) and BAX (**D**) in MDA-MB-468 (p53^R273H^) or MDA-MB-231 (p53^R280K^) cells treated with DPEITC, CPT, or both for 24 h were analyzed by qRT-PCR. The *p*-values are as indicated (*** *p* ≤ 0.0005, ** *p* ≤ 0.005 and * *p* ≤ 0.05). Experiments were performed in triplicate. Error bars represent the SD.

**Figure 9 cancers-15-00928-f009:**
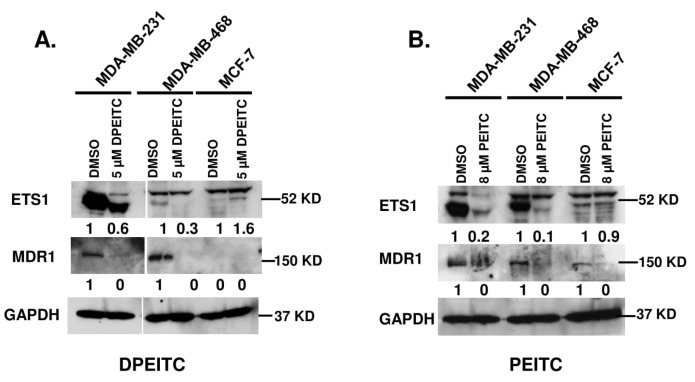
DPEITC reduces the expression level of ETS1 and MDR1 in mutant p53 TNBC cell lines. MCF7 (WT p53), MDA-MB-231 (p53^R280K^), and MDA-MB-468 (p53^R273H^) were treated with 5 μM DPEITC (**A**) or 8 μM PEITC (**B**) for 24 h. Control cells were treated with DMSO. The cell lysates were resolved by SDS-PAGE, probed with anti-ETS1 and anti-MDR1 antibodies, and re-probed with anti-GAPDH antibody. Experiments were performed in triplicate.

## Data Availability

The data presented in this study are available in this article.

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
