# Peer review of "2,2-Diphenethyl Isothiocyanate Enhances Topoisomerase Inhibitor-Induced Cell Death and Suppresses Multi-Drug Resistance 1 in Breast Cancer Cells"

_cancers, 2023, doi:10.3390/cancers15030928_

Round 1

Reviewer 1 Report (Previous Reviewer 1)

The manuscript Monika Aggarval is a revision and the author has addressed the comments of the Reviewer satisfactorily.  However, the new version of the manuscript requires minor changes of the english language used in the new additions to the text before it can be accepted for publication.

Author Response

We would like to thank the Reviewer for the comments. In this revised manuscript we have made changes as suggested by the Reviewer. The following is our detailed point-by-point response.

The manuscript Monika Aggarval is a revision and the author has addressed the comments of the Reviewer satisfactorily.  However, the new version of the manuscript requires minor changes of the english language used in the new additions to the text before it can be accepted for publication.

Response. We have our manuscript read by Dr. Robert M. Brosh (see Acknowledgements).

Reviewer 2 Report (Previous Reviewer 2)

Thank you for responding to all comments. The manuscript is ready for publication in its present form. 

Author Response

We thank the Reviewer for accepting our manuscript.

Reviewer 3 Report (Previous Reviewer 3)

Having previously read this manuscript, I can already tell that it has undergone a significant improvement. The author further supports the rationale and hypothesis with several new experiments. I still have a few minor concerns:

1.      It has been demonstrated that DPEITC causes apoptosis, and the flow cytometry data for apoptosis are shown in supplementary figure 2. I'm not sure if the flow cytometry demonstrates Annexin binding or not because the figure doesn't make it very clear. To comprehend it better, it might need to be explained in detail.

2.      Clarify whether the western-blot image for p53 (FL393) serves as an input control in the supplementary figure 4A. The immunoprecipitation figure's methods and description can be made better.

3.      To further support the conclusions, some of the supplementary figures should be incorporated into the main figure.

Author Response

We would like to thank the Reviewer for the comments. In this revised manuscript we have made changes as suggested by the Reviewer. The following is our detailed point-by-point response.

Having previously read this manuscript, I can already tell that it has undergone a significant improvement.

Response. We thank the Reviewer for appreciating our efforts.

1. It has been demonstrated that DPEITC causes apoptosis, and the flow cytometry data for apoptosis are shown in supplementary figure 2. I'm not sure if the flow cytometry demonstrates Annexin binding or not because the figure doesn't make it very clear. To comprehend it better, it might need to be explained in detail.

Response: We thank for the suggestion. We have added more information in the materials and methods for Annexin V staining (see p. 8), the result section (see p. 12, 13 and 17), and labeling in the supplementary Figure S2 and S7B in the revised manuscript.

2. Clarify whether the western-blot image for p53 (FL393) serves as an input control in the supplementary figure 4A. The immunoprecipitation figure's methods and description can be made better.

Response.  We have now added methodology for immunoprecipitation under materials and methods (see p. 10), more details in results (see p. 14), and in figure legend (see p. 33 Figure 2A). 

 3. To further support the conclusions, some of the supplementary figures should be incorporated into the main figure.

Response. We thank for the suggestion. We have now moved Supplementary Figure S4A, Supplementary Figure S6, and Supplementary Figure S8 to the main text in the revised manuscript as Figure 2A, Figure 3, and Figure 4, respectively. We have revised the figure and supplementary figure numbers for the rest of the manuscript accordingly.

We have also added representative cell cycle histogram images for results shown in Figure 8A in the revised manuscript (see p. 20 and Figure S10).

We have now improved the methods and the results, as needed.

Round 2

Reviewer 1 Report (Previous Reviewer 1)

The authors addressed all the comments of the reviewers satisfactorily.

Reviewer 3 Report (Previous Reviewer 3)

Significant revisions have been made to the manuscript. I don't have any more questions.

This manuscript is a resubmission of an earlier submission. The following is a list of the peer review reports and author responses from that submission.

Round 1

Reviewer 1 Report

The manuscript by Monika Aggarwal aims at demonstrating that DPEITC, a synthetic ITC, is a  powerful anti-neoplastic agent in TNBC breast cancer cell lines characterized by p53 mutations.  The conclusions are based on the use of 4 cell lines and are definitely overstretched relative to the results presented.  The study should be conducted on a larger number of cell lines which must include TNBC lines with no p53 mutations.  Overall, the study is designed rather poorly and the presentation of the results is confused and totally unsatisfactory. Finally the synergistic effects of DPEITC and doxorubicin or camptothecin may be of therapeutic interest.  However, the synergism should be evaluated in TNBC cell lines devoid of p53 mutations to support the suggested mechanisms of action.

Reviewer 2 Report

For more than twenty years, different groups have studied the role of Isothiocyanates, such as Phenethyl isothiocyanate, PEITC and its effect on the hotspot’s residues of P53 mutants and as an anticancer drug. Also was reported before that the more the increase in the lipophilicity of arylalkyl ITCs by adding an aromatic ring, the more the potency of the isothiocyanate molecules. Due to this notion, it was shown that using 2,2 diphenyethyl isothiocyanate, DPEITC is more potent than PEITC in depleting P53 mutants and is a more potent inducer of apoptosis than many isothiocyanates including PEITC.

Based on the above, the author of the current manuscript carried out a study to describe the effect of an additional aromatic ring on the potency of DPEITC and its ability to target P53 mutants alone or in combination with chemotherapeutic drugs on breast cancer.

I want to mention the following:

1.       On page 2, the author stated, “our previous structural-activity relationship (SARs) study…………… “and quoted reference 18. When I checked reference 18, I could not find the author of this manuscript in the author’s name of Ref 18. I also found a slight difference between the names mentioned in ref 18 and the actual publication.

2.       The source where DPEITC is obtained has not been mentioned in the manuscript.  

3.       on page 5, the author stated that “ -------The upregulation of MDM2expression was detected in ------------upon treatment with 5um DPEITC, suggesting that DPEITC can rescue mutant P53 and also activate WT P53. MDM2 is a negative regulator of P53, and its upregulation will negatively affect the activity of P53.  How the upregulation of MDM2 expression could do otherwise, as the manuscript mentioned?

4.       In Page 12, paragraph, “---------previously, we showed that ITCc can (should be could) bind the cysteine residue in DNA binding ---------” there is no reference to this previous study.

5.       The author showed that DPEITC could rescue the mutant P53 and does not affect the WT P53. This may be due to the folding of the P53 mutant. I wonder what these molecules could do to the wild types of proteins in the cell.  Can the author comment on this, please?

6.       Many studies from the last 15-20 years show the anticancer effect of the ITCs through P53 mutants. I wonder if it is that important why there is no significant work on humans and some clinical trials to show the effect in vivo.

The manuscript contains a significant amount of work, similar to previous work on other ITCs. It, however, reported for the first time the synergy effect of ITCs with a chemotherapy drug via mutant P53 reuse.

I propose the publication of the manuscript after dealing with the above points.           

Reviewer 3 Report

This manuscript is focused on a highly important and relevant topic, in which the author demonstrated that the synthetic analog DPEITC selectively targets mutant-p53 in breast cancer regardless of the p53 mutant type. Furthermore, topoisomerase inhibitors and DPEITC have been shown to work synergistically to inhibit the growth of breast cancer.

The majority of human cancers have been shown to have mutations in the tumor suppressor gene p53, which results in the unregulated expression of several downstream molecules that aid in cancer proliferation and metastasis. Exploring inhibitors that selectively target mutant-p53 is one of the hot topics being researched by several groups, with limited success so far. This article makes a novel therapeutic argument for DPEITC's use in pursuing p53-mutated cancers.

There are some points that need to be addressed:

1.     In the manuscript, the author used the term ‘restored mutant-p53’, which seems paradoxical to me as the mutant-p53 normally promotes tumorigenesis, and restoring them may indicate their increased activity. I believe the author meant to change mutant-p53 activity to WT-p53 activity, and some relevant terminology may help clear up the confusion.

2.     The introduction section, in my opinion, should be more effective and shorter.

3.  The catalog numbers for the majority of the reagents and chemicals used are missing and should be included in the manuscript.

4.    '2.6 Flow-cytometric analysis' in the material and method section can be changed to 'Cell cycle analysis' because it primarily describes the cell cycle analysis. Flow cytometry has also been used in other experiments.

5.     To confirm DPEITC's selective targeting of mutant-p53, inhibitors that stabilize p53 expression, such as Nutlin-3a, can be used to demonstrate the rescue effect on p53 decrease.

6.     It can also be further shown by using p53-specific siRNA/shRNA to confirm the selective targeting of mutant-p53 by DPEITC.

7.   Representative flow-cytometry images for Annexin V staining may provide a clear picture of apoptosis following DPEITC treatment.

8.     It has been reported that once the anti-apoptotic Bcl-2 proteins are neutralized, p53 is no longer required for Bax activation; therefore, the author should evaluate further to see if this is simply due to mutant-p53 inhibition by DPEITC.

9.     Several factors influence the cell cycle by promoting cellular stress. How does the author explain that it is not the result of stress caused by DPEITC treatment, but rather of DPEITC specifically suppressing mutant-p53?

10.  A representative cell cycle analysis histogram image may provide a clear picture of the effect.

11.  Why are the smaller differences (DMSO vs DPEITC) more significant than the larger ones in figure 2A?

12.  Page 9; Section 3.7: MDA-MB-468 cells are shown to have p53R280K mutation. Is it a typographical error or exogenously expressed mutant-p53?

13.  For clinical relevance, the findings should be validated using animal models.

14.  I couldn't find any data that directly emphasized the mechanism of DPEITC-mediated regulation of mutant-p53. Can the author demonstrate experimentally how DPEITC directly affects mutant-p53 expression?
